# Understanding the Family Context: A Qualitative Descriptive Study of Parent and NICU Clinician Experiences and Perspectives

**DOI:** 10.3390/children10050896

**Published:** 2023-05-17

**Authors:** Maya Dahan, Leahora Rotteau, Shelley Higazi, Ophelia Kwayke, Giselle Lai, Wendy Moulsdale, Lisa Sampson, Jennifer Stannard, Paige Terrien Church, Karel O’Brien

**Affiliations:** 1Department of Pediatrics, Division of Neonatal-Perinatal Medicine, University of Toronto, Toronto, ON M5S, Canada; 2DAN Women & Babies Program, Department of Newborn and Developmental Pediatrics, Sunnybrook Health Sciences Centre, Toronto, ON M4N 3M5, Canada; 3Centre for Quality and Patient Safety, University of Toronto, Toronto, ON M5S, Canada; 4Department of Pediatrics, Mount Sinai Hospital, Sinai Health System, Toronto, ON M5G 1X5, Canada

**Keywords:** communication, neonatology, family context, decision making

## Abstract

Enabling individualized decision-making for patients requires an understanding of the family context (FC) by healthcare providers. The FC is everything that makes the family unique, from their names, preferred pronouns, family structure, cultural or religious beliefs, and family values. While there is an array of approaches for individual clinicians to incorporate the FC into practice, there is a paucity of literature guiding the process of collecting and integrating the FC into clinical care by multidisciplinary interprofessional teams. The purpose of this qualitative study is to explore the experience of families and Neonatal Intensive Care Unit (NICU) clinicians with information sharing around the FC. Our findings illustrate that there are parallel and overlapping experiences of sharing the FC for families and clinicians. Both groups describe the positive impact of sharing the FC on building and sustaining relationships and on personalization of care and personhood. The experience by families of revolving clinicians and the risks of miscommunication about the FC were noted as challenges to sharing the FC. Parents described the desire to control the narrative about their FC, while clinicians described seeking equal access to the FC to support the family in the best way possible related to their clinical role. Our study highlights how the quality of care is positively impacted by clinicians’ appreciation of the FC and the complex relationship between a large multidisciplinary interprofessional team and the family in an intensive care unit, while also highlighting the difficulties in its practical application. Knowledge learned can be utilized to inform the development of processes to improve communication between families and clinicians.

## 1. Introduction

In the late 20th century, decision-making in neonatology and in the wider medical field shifted from a paternalistic model, in which physicians make decisions regarding care for patients and families, towards a model guided by recognition of patient autonomy [1,2]. The latter model established a professional standard of providing detailed information about possible outcomes with the goal of allowing families to make independent ‘informed decisions’ [3,4]. However, research in psychology and behavioural economics has shown that data-guided parental choice is not categorical and bias-free [3,5]. Cognitive biases (such as framing effects, availability biases, commission bias, etc.) are omnipresent, especially in conversations with families [5,6,7,8]. Research has also highlighted that individuals make decisions based on their own lived experiences and values [9,10,11,12]. The pendulum has therefore swung back to the middle, to the intermediary between the decision-making spectrum extremes: a shared decision-making (SDM) model [12,13,14,15]. The SDM model, however, is not without its critics [16,17]. While the ideal model may remain debated, the need to understand a family’s context (FC) is well agreed upon to be a crucial first step in building a relationship to facilitate decision-making with families [12,18,19]. The FC is everything that makes the family unique, from their names, preferred pronouns, family structure, and cultural or religious beliefs, to family values.

Positive impacts of learning about and sharing the FC have included facilitating clinicians’ ability to have a holistic mindset [20] and to personalize care [12]. Contextualization of care, defined as adapting the medical plan to the patient’s life context, or in the case of a neonate, the family’s context, is a key competency for physicians [21,22]. There is an array of approaches for the individual clinician, often the physician, to collect and incorporate the FC into practice [12,18,23,24,25]; yet, there is no literature guiding the process of collecting and integrating FC into clinical care provided by a multidisciplinary interprofessional team.

The purpose of this descriptive qualitative study is to explore the experience of families and Neonatal Intensive Care Unit (NICU) clinicians with information sharing around the FC. We aim to further the understanding within the literature of how the quality of care is impacted by clinicians’ appreciation of the FC and the complex relationship between a large multidisciplinary interprofessional team and the family in an intensive care unit setting. Understanding this issue from the perspective of families and clinicians is the critical first step to improving care [26,27]. Knowledge gained with this practice may then be utilized to inform the development of quality improvement (QI) processes aimed at improving communication between families and clinicians.

## 2. Methods

We conducted a descriptive qualitative study in a 42-bed tertiary care neonatal unit in Toronto, Canada, that cares for inborn and outborn infants requiring tertiary neonatal care. Patients reflect the broad multicultural and multilingual diversity of the City of Toronto.

Using purposive sampling, we sought maximum variation by selecting families with varying cultural backgrounds and educational levels and clinicians with varying professions, cultural backgrounds, and experience, were recruited [28]. Recruitment was stopped for each group when thematic saturation was reached [29].

This study was reviewed and approved by the Research Ethics Board of Sunnybrook Health Sciences Center. Each participant consented to participation. Semi-structured interviews were used to collect similar information from the participants while allowing personal stories and new concepts to emerge [30]. They occurred between August and December 2021. Interviews with the families were conducted by a physician (MD), and interviews with the clinicians were conducted by a research assistant with no prior relationship to the NICU or its staff. Semi-structured interview guides were used to lead the interviews and were modified in response to an iterative analysis. The development of the guides was informed by a review of the literature. The guides were piloted with an unrelated participant. They broadly focused on what participants shared about the FC, their experience of sharing the FC, and the decisions around how and when to share information about the FC [31]. The guides are available in Appendix A and Appendix B [31]. Interviews were transcribed using artificial intelligence and corrected and anonymized by the lead researcher (MD).

A thematic analysis was performed on the interview transcripts [32]. Two authors (MD + LR) read the first four transcripts for both groups (families and clinicians) and performed a preliminary analysis to generate a coding structure. The coding structure was then used by MD to code the remainder of the transcripts. The two lead researchers held regular analytic meetings following the steps for thematic analysis outlined by Braun and Clarke [33]. Through coding and analysis, we identified, defined, and named two important themes related to the experience of sharing the FC: the process of sharing the FC and the impact of sharing the FC. The two themes were examined for how they were similar and different among and between the parents and clinician groups.

The lead researcher (MD) was a neonatal fellow working at the institution. Although there was a degree of involvement in the care of the patients whose families were interviewed during on-call shifts, MD was not a member of their core clinical team. The second researcher (LR) held solely a research role and was not part of the neonatal care team. The remainder of the research team was part of the multidisciplinary interprofessional team at the research site. The research team helped formulate the project details and provided feedback on the interview guide as well as a review of this manuscript.

## 3. Results

### 3.1. Demographic of Participants

Eleven parents making up eight families were interviewed (Table 1). One parent removed themselves from the study after the interview was completed; their interview was not included in the analysis. All parents had infants who were born at less than 29 weeks of gestation and with birthweights less than 1000 g. Infants were 3–14 weeks old at the time of the interviews. Four of the ten participants’ first language was not English. Eleven clinicians from varying disciplines were interviewed (Table 2).

### 3.2. Thematic Analysis

Two interconnected overarching themes were identified in both the family and clinician interviews: *the process* and *the impact* of sharing information about a family’s context (FC). Figure 1 shows the parallel between the two themes and their subthemes as experienced by both families and clinicians.

#### 3.2.1. Process

The first theme describes the process by which families shared information about their FC with clinicians and the process by which clinicians shared information about FC with the other members of the care team.

#### 3.2.2. The Process: Family Perspective

In the following section, the family perspective of the process is presented using two subthemes: *revolving clinicians*, which describes the process of information sharing as seen by families and how they experienced it, and *controlling the narrative*, describing parents’ underlying desire to control the narrative about their context.

### 3.3. Revolving Clinicians

Parents described a general openness to sharing information about themselves and their family, which, when asked, usually occurred informally at the bedside. When asked whom they shared information with, families invariably brought up the seemingly endless revolving rotation of clinicians. George explains:

“*There’s people coming in and out all the time, and you have one doctor on Saturday morning, and you never see them again, there’s another doctor on Saturday night, you never see them again […] you develop a sort of little relationship with someone and then you never see them again*.”

With this perpetual change in providers, parents noticed a disconnection between providers related to the FC. George continued to describe the need to create *“an efficient narrative, otherwise you’re just exhausted*” because of the constant need to reiterate this narrative to each new provider. Chandan emphasized this constant repetition “*if we had the consistency, then they already knew us. And they knew that, you know, they don’t have to provide with the same information each and every day*”. Chloe worried that one day with “*fatigue setting in, we’re just not going to give all the info […] and then [the clinicians] won’t get the whole story and they wouldn’t be able to cater their treatment to our situation*”. The need for repetition was specific to the family context and did not include the infant’s medical history. Parents described their experience of sharing information about their context as generally positive with individual clinicians but exhausting due to the repetition.

### 3.4. Controlling the Narrative

Parents had an underlying desire to control the narrative about their context, both its creation and its evolution. For example, Chloe preferred to be asked upfront questions about her context because it would make her “*feel like I would have control over the situation a little bit more*”, explaining she already felt a lack of control in their NICU journey. She was also initially skeptical about sharing information about her context, unclear about the “*tenor of the interaction […] maybe it’s a question of utility, like what is that information used for*”. Simi wanted clinicians to understand her busy home life with her four school-aged children. Similarly, Shaden felt “*it helps reduce the guilt when I have to leave […] it’s so much easier if people just know that bigger context, right?”* once the bedside nurses knew she picked up her son from school every day. Several of the parents described forging relationships with those caring for their child as they believed they would get even better care if clinicians “*remember [them], you know, just trying to keep [baby] on the map, even though I know it’s their job. But just to me, it’s like the one thing I can do is like, engage them in that way*” (George).

Shaden initially feared being stigmatized because of her appearance: “*I know that systemic biases exist, and don’t know how it would necessarily play out here. That gap makes it scarier”*. As she was able to share more about her context and feel more understood, she described the fear dissipating. Her ability to share her FC increased her sense of personhood and reduced her feelings of stigmatization. Overall, families desired a sense of control over one of the few variables that was within reach when their infant is unexpectedly hospitalized in the NICU: their own family narrative.

#### The Process: Clinicians Perspective

In the following section, the process is described from the clinicians’ perspective. When discussing the process for sharing information about the FC with clinicians, participants noted the importance of both sharing between families and clinicians and between clinicians. Two subthemes are outlined: *the broken telephone* and *controlling the narrative*.

### 3.5. The Broken Telephone

Clinicians described relying on verbal handover due to the inconsistency in written documentation about the FC. Though several clinicians commented on the potential risks of relying on verbal information sharing; “*if it’s not written, it didn’t happen. And that’s how stories get made up*” (RN), they rarely reported looking at the patient chart to find this information. Finding information about the FC was difficult, due to sporadic documentation and the many possible locations to document within the electronic medical record (EMR). Throughout the interviews, 17 different locations were identified where information about the FC could be found, many of which were discipline-specific. A social worker, who regularly documented in the chart, expressed her frustration with this approach; “*people don’t always read the chart […] and so it all has to be repeated […] that’s not efficient, and it doesn’t feel all that respectful”*.

Poorly written handover practices about the FC and the subsequent reliance on verbal handover created fertile ground for information disconnection described as a broken telephone. One nurse described this process: “*It’s kind of like if somebody hears a story […] especially juicy stories […] a lot of it gets embellished as the story goes down the telephone line. So maybe things are kind of, exaggerated*”. Another nurse elaborated: “*What would happen is a rumor mill. Okay, it’s not necessarily written down, but what people say may not be true, or they may have a different opinion or they’ve misunderstood something*”. A neonatologist gave examples of how this approach left room for biases:

“*You’re making assumptions about people based on something about their background. Teenage mothers, people who take drugs, cultural things sometimes, you know, so there’s different reasons that I think we’re either labeling or judging those sorts of things. That’s not really positive*.”

The ‘broken telephone’ was identified as a serious gap in their ability to care for patients, with potential detrimental effects on families.

### 3.6. Controlling the Narrative

Clinicians wanted equal access to the FC in order to better care for patients since “*the social and the medical needs to go together in order to better serve [families]*” (RN). However, many clinicians reported inconsistent access. A social worker explained how everyone wants to contribute to it in their own way, independent of their professional role:

“*There are some practitioners, regardless of discipline, who make a point of trying to really learn about the family. And it doesn’t mean that they’re necessarily the ones sitting down and asking the family about their circumstances about their supports about mental health history about their understanding of the medical, you know, they’re gathering it from colleagues, from other people’s interactions with the family, from documentation in the chart*.”

Each clinician articulated a goal to create their own relationship with the family but also an interest in furthering the relationship between the family and care team. In addition to providing better care, strong relationships also improved their own work experience. This desire to understand the FC was heightened in the more medically fragile infant. “*If the baby’s critical […], you might speak to them every day*”, said one of the neonatologists as they described the increased volume of interactions and the perception of a more imminent impact on the family and the baby. Clinicians did not seek to control the content of the family narrative the same way parents did; instead, they sought equal access to it to be able to support the family in the best way possible related to their clinical role.

## 4. The Impact

The second overarching theme describes the impact of sharing information about the FC for clinicians and families; specifically, the *impact on relationships* and the *respect of personhood*.

### 4.1. The Impact: Families Perspective

#### 4.1.1. Impact on Relationships

Parents felt a connection and establishment of trust when clinicians demonstrated interest in the FC. They spoke of “*highly personalized information catered to us*” (Chloe) when the FC was well understood. As described above, parents experienced an improvement in care once they established relationships with clinicians. Shaden explained:

“*It was so helpful because then everybody knew that we had another kid […] It was a game changer. Like we got to do kangaroo care* (Kangaroo care is a method of holding a baby skin to skin) *so much more and I think it was just helpful for people to know that there was this other major piece that governed our interactions*.”

She continued to elaborate on how it made a big difference when a nurse knew her context; it “*builds trust, like you really need that with this situation. Like, if you can’t have a communication I don’t know how you’d be able to leave your baby in the room and walk away*”. Marie spoke about how the ability to bond over commonality helped lessen her anxiety; “*the nurse basically said her son went through it and he played he started playing after surgery right after the next day so […] okay, it’s not gonna be that bad to do the surgery*”. Sarah described the emotional relief provided by an opportunity to speak about something other than her baby’s day to day care: “*a little off the topic [but] I’m able to share excitement about my job […] so I felt like it was it was good for me […] even though the focus is on [baby] it gives me that break from my head”*. For Sanjeeva and Chandan, sharing their FC was the root of their relationships:

“*It makes us feel more connected. And then we can share the thoughts with the nurses. And then that way, we develop a connection with the nurses as well, because they also understand where we come from, what we do, and then I would say, it develops in affection as well with the family*.”

However, not every parent spontaneously shared their FC. Zain and Bianca, self-described as quiet and introverted people, worried they would be disliked if they overshared. They only shared when asked. When the door was opened for them to share their beliefs and values, it allowed for services to be provided that they otherwise would not have known about and also made them “*feel good that somebody wants to know what my beliefs are*” (Bianca).

#### 4.1.2. Impact on Personhood

Taking the time to understand the FC was interpreted by parents as respecting their personhood. Parents spoke about the drastic turn their life took when they had a preterm child and appreciated when clinicians took the time to understand who they were as individuals. It made parents feel like the clinicians “*care about both my son and myself*” (Simi). Sarah elaborated on why this is important: “*Mom also needs to heal as much as baby needs to heal. So it’s a good healing process for mom*”. Parents gave examples of how this respect for their personhood was displayed. For Bianca, it was the meticulous care for her necklace she had left in her baby’s incubator and the books she had placed in the room; for Marie, it was showing interest in her business and getting updates on her family members; for Shaden, it was checking in about her mental health; for Zain, it was being offered the opportunity to share his religious and cultural views; and for George and Chloe, it was the opportunity to talk about their family values.

Taking the time to understand the FC fostered trust in the individual clinician but also had the ability to positively or negatively colour the relationship with the entire group represented by that single clinician (e.g., professional group, the unit within the hospital, etc.). George compared their experience in the NICU with their experience in the high-risk obstetrical unit despite only having met a few of the physicians in each unit:

“*The doctors in the NICU have been more warm and more generous with their time than the doctors and OBs that we encountered in the floor up, the sort of labor birthing floor, I forget the names of those areas, but they were more sort of business. And sometimes they would just sort of talk to each other and ignore us, you know? And we’re like, hey, like, we have a question or what’s going on? Or what are you talking about? Or why is there a worry tone in your voice? And that’s just not the case, in NICU, we’ve loved all the doctors there*.”

Chandan compared primary nurses to all other nurses:

“*I feel if they are primary, they feel like they own it, if they are not primary then why bother to know the baby or, it’s more of an attitude with the baby. I don’t have to know the baby, I am just here, I just have to take care of them and go*.” 

Bianca compared her experience in the NICU to her previous medical encounters, highlighting how being asked about her context was seen as a sign of respect: “*there are some doctors that don’t even let you talk. Here, they ask what your culture is*”. However, she also reflected on how she initially hesitated to share, fearful of judgement based on her visible demographics. Hearing stories of “*some of the anti-black racism [or] the Islamophobia […] especially when you’re in such a vulnerable position, you’re […] worried that that’s going to play a role”*. All of these examples speak to the nuances of how communication shaped the feeling of respect that is reflected upon an entire care team, as well as the hesitation parents may have to share.

### 4.2. The Impact: Clinicians Perspective

#### 4.2.1. Personalization of Care

Clinicians overwhelmingly spoke about the positive impacts of understanding the FC, such as the increased ability to be respectful, empathetic, and accepting, allowing for a greater appreciation of the individuality of each family and a better ability to personalize care. A respiratory therapist (RT) spoke about discovering “*those special precious moments that they’ll never get back*” that is of unique importance to certain families, but clinicians will only know if they ask. Another clinician elaborated: “*Some religions or cultures have certain practices that are time sensitive […] so if we know what those are, we can help identify and have the resources available to us and to the families*” (NP). This is crucial in the care provided as “*the patient […] is not alone. They exist within a family, a framework. And family doesn’t just mean mum and dad, it can mean you know, a broader community*” (MD). Understanding the FC not only helped personalize what was offered to parents but also how clinicians approached communication.

“*If a mom was raised with a sibling who had cerebral palsy, they are worried about different things, then a family who’s got no medical background, and has never had a baby in the hospital. So their stressors are different. And how we help them cope can sometimes be different*.”(NP)

This personalization afforded opportunities to adapt the approach to the individual family, anticipating and addressing their unique concerns.

“*When we understand these things about families and their facilitators and barriers to being at the bedside and helping their children grow and develop, which we know is important […], then we’re better able to help them be present at the bedside*.”(NP)

Sharing this type of information between clinicians helped “*the next person that’s meeting the family [to] have a little bit more context as to where they’re coming from and, and what might be important for them*” (NP). Ultimately, “*the more we know about the person, the family, them in their environment, them within this medical environment, what their needs are, their questions, I think, the better able we are to provide the care that we do*” (SW). Clinicians unanimously viewed this shared mental model as facilitating better care.

#### 4.2.2. Forging Relationships

Beyond the ability to provide personalized and empathetic care for babies and families, understanding the FC also provided an opportunity for clinicians to forge relationships with the families. A nurse practitioner described that there is “*something that’s kind of nice to be able be able to have a conversation with parents*” (NP) regarding life outside the NICU. It improved the sense of collaboration clinicians feel with families (MD), and all clinician participants reflected positively on their experience of caring for families with whom they were able to further their relationship by understanding their FC. However, in the context of the busy NICU environment, several clinicians reflected upon concerns regarding the time commitment to collect and share information on the FC.

## 5. Discussion

The present study described how families and clinicians experienced sharing information about the FC. The impact of knowing and sharing this information was positively described by all participants, but the process was highlighted as fragmented. The benefits of sharing the FC have been described across a range of settings including adult, pediatric, and neonatal units [6,12,18,20,21,24]. Our findings complement the existing literature by further exploring these impacts, specifically in the NICU setting. Sharing FC supports relationship building and personalization of care and provides an opportunity to respect the personhood of patients and families [34].

Study findings described how both clinicians and families identified a desire to control information about the FC. Control was perceived as enabling enhanced patient-centred care. However, families and clinicians seemed to have different motivations for controlling the narrative. Families wanted control over *shaping* their narrative and how they were perceived, while clinicians wanted ease of *access* to the narrative to better support their patients. A similar theme was found in the literature under the umbrella of narrative competence. Narrative competence is defined as “the set of skills required to recognize, absorb, interpret, and be moved by the stories one hears or reads” [35]. Narrative competence helps clinicians and families engage with each other not only in the patient–clinician relationship but also in the human–human relationship, improving the experience for all those involved [36,37]. The theme described by our participants showed a desire to use the family context as a means of building a relationship between parents and clinicians to improve care.

While families and clinicians alike discussed this shared goal in the interviews, they also spoke about the skills and the infrastructure needed to facilitate narrative competence in the NICU setting [34]. Key to achieving the goal of relationship building is *the process* by which information about the FC gets shared by families and between clinicians and the perceived enablers and barriers. Parents and clinicians described how the process varied between individual clinicians and referred to the broken telephone and fragmentation that occurred due to the revolving door of clinicians involved in each infant’s care throughout their long journey in the NICU.

Beyond the lack of infrastructure to support sharing the FC, participants in our study also alluded to the perceived pitfalls of integrating FC into care. Clinicians worry about the time required to understand the FC. Some parents in our study hesitated to share their FC because of the fear of being stigmatized based on their context. Concerns in the literature also centred on the time necessary to understand a patient’s context, the education, and the culture shift necessary for its success [38,39]. Interestingly, while people worry about being stigmatized, it is argued that the antidote to this stigma, founded in implicit bias, is actually contextualizing care [40].

Our findings highlight the variability in clinician communication skills and documentation, leading to the reliance on verbal handover, which is fraught with the risk of inaccuracies and misinterpretation. Though certain clinicians felt that understanding the FC would help them provide more narratively competent care, they did not know whether this information was ever collected by their colleagues, or if so, where to find it within the patient chart. We found that sharing the FC often relied on verbal handover. Similar to unstructured verbal handover of clinical information, reliance on verbal handover of FC risks the omission of important details, unnecessary inclusion of superfluous information, and the ‘broken telephone’ phenomenon [41]. Through the interviews, we discovered that information about the FC was being documented in 17 different places within the medical record. Participants described how this inconsistency can lead to miscommunication and gossip, requiring families to repeat themselves or correct misinformation. Clinician participants also described the complexities of handover within a large multidisciplinary interprofessional team, highlighting an area for improvement in the care of neonates and their families.

Fragmentation of communication is a problem that has been described before in terms of relationships and experiences of care [41], which can be addressed with the standardization of various processes across care providers, development of guidelines, communication tools, standardized order sets, and checklists [42,43,44,45,46,47]. Similar approaches can potentially be applied and adapted to communication about the FC but require further exploration to ensure an appropriate balance between the standardization of the approach to collection and documentation while still promoting the personalization of information.

Despite these technical challenges highlighted by participants, findings from the interviews for both clinicians and families highlight the importance of consistency in care providers during the often lengthy NICU stay. Consistency led to a better understanding of the FC, which in turn resulted in building trusting relationships; it is these relationships that lessened parental anxiety.

Our study has limitations. Firstly, only parents who felt comfortable being interviewed in English were included. While families that did not speak English well enough to participate in an interview are likely to have had a different experience based on the ease of communication, English was a second language for almost half of our participants. Secondly, we only interviewed families and clinicians about their experiences, we did not observe them interacting. Additionally, this was a single-site study and focused on the NICU. While a focus on the uniqueness of the NICU setting allowed further depth and a better understanding of the intricate dynamics that exist within the team, this may limit the generalizability to other settings. A strength of our study was that we interviewed clinicians with varying backgrounds professionally and culturally. Moreover, the use of qualitative methods facilitated a deeper exploration of the process the team used to understand the FC as well as the nuances in the impact of knowing the family context had on the care provided.

## 6. Conclusions

Results of this descriptive qualitative study emphasize the vastly positive impacts of sharing the FC on the care provided to infants and their families in the NICU, while also highlighting the difficulties in the practical application of this practice. It furthers the understanding within the literature of the impact of integrating the FC on care experiences and the complex relationship between a large, multidisciplinary, interprofessional team and the family in an intensive care unit. Armed with this knowledge, a targeted approach can be created to improve the current process by addressing gaps highlighted by families and clinicians and focusing on the positive impacts described. Beyond the implementation of a tool or a checklist, family context needs to be integrated into each aspect of clinical care to facilitate the narrative competence desired by families and clinicians alike.

## Figures and Tables

**Figure 1 children-10-00896-f001:**
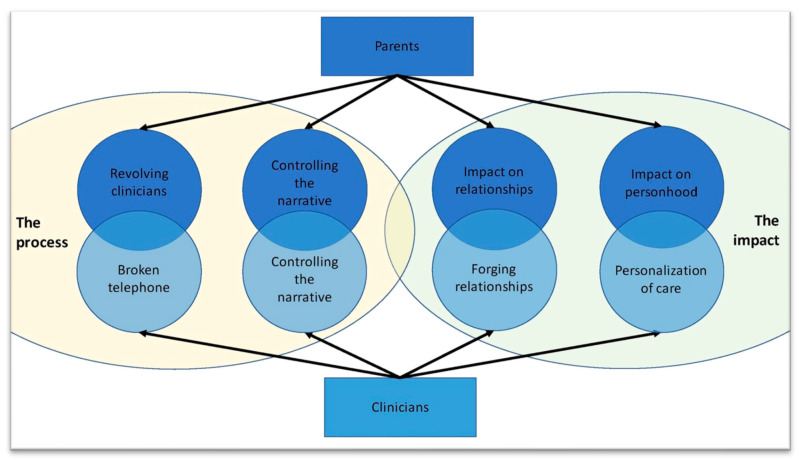
Venn diagrams visually depicting the relationship between two overarching themes and their subthemes. It highlights the parallels and overlapping experiences of the families with those of the clinicians.

**Table 1 children-10-00896-t001:** Demographics of Families.

Fictitious Name	Gender	Marital Status	Other Children	Self-Identified Ethnicity	English as Mother Tongue	Highest Level of Education	Child’s Gestational Age *
Chloe	Female	Married	None	White	Yes	PhD	26–27 weeks
George	Male	Married	None	White	Yes	PhD	26–27 weeks
Sanjeeva	Male	Married	None	South Asian	No	University	24–25 weeks
Chandan	Female	Married	None	South Asian	No	University	24–25 weeks
Zain	Male	Common law	None	Black	No	Trade certificate	24–25 weeks
Bianca	Female	Common law	None	Black	Yes	High school	24–25 weeks
Simi	Female	Married	2+ other children	Black	Yes	College	28–29 weeks
Marie	Female	Single	None	Black	Yes	University diploma	22–23 weeks
Shaden	Female	Married	1 other child	Arab, South Asian	Yes	University—above bachelor	22–23 weeks
Sarah	Female	Common law	None	Black	No	Bachelor’s degree	26–27 weeks

* GA reported as a range to preserve anonymity.

**Table 2 children-10-00896-t002:** Demographics of Clinicians.

Discipline	Number of Participants
Neonatologist	3
Nursing team leader	3
Nurse practitioner	3
Social worker	1
Respiratory therapist	1

## Data Availability

The data presented in this study are available on request from the corresponding author. The data are not publicly available due to it being qualitative data and can breach patient confidentiality.

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
