# Peer review of "Understanding the Family Context: A Qualitative Descriptive Study of Parent and NICU Clinician Experiences and Perspectives"

_children, 2023, doi:10.3390/children10050896_

Round 1

Reviewer 1 Report

The idea of the article is of undoubted interest, the topic is relevant and timely. A big ball of problems that has accumulated around important information from the child's family context (FC), which is rarely adequately owned by medical professionals, did not begin to form today. It includes several aspects.

On the one hand, indeed, doctors at NICU work in shifts, respectively, the children who get there are seen by a different doctor every day. This is a given that is almost impossible to change in the existing system of medical care. On the other hand, the history of a child's life is rarely described in the medical history in detail and with a clear fixation of FC. This skill is formed at the stage of both undergraduate and postgraduate education of physicians working with children. This can probably be improved through the introduction of appropriate training, if it is clearly defined what to strive for. On the third hand, the parents of today's children are very different from those with whom we worked 30, 20 or 10 years ago (the theory of generations). Modern parents are distinguished by the peculiarities of thinking (the so-called "clip thinking"), which should also be taken into account when developing training programs for those working at NICU about how and where to fix FC in the MR. Thus, the relevance and timeliness of the topic are undeniable.

However, the format of the study (qualitative descriptive study) does not imply obtaining highly accurate information expressed in numbers and degree of significancy, which makes it impossible to apply the results obtained in practice. This is just a description of the problem. And that's all. I, as a reviewer, lacked a description of the logic of the selection of respondents. On what basis were they selected - those who simply agreed to answer? For a general statement of the problem, this is enough. In order to rethink how to improve this situation, no. Perhaps it made sense to form several identical groups, uniting the respondents according to racial, ethnic, linguistic, psychotypical and other characteristics.

Thus, it seems to me that the study, which is certainly important and necessary, should be continued so that it becomes possible to make reasonable proposals on how the situation can be improved.

From minor remarks, the current approach to decision-making (patient-oriented), which has replaced the patternalist one, is characteristic of all pediatrics (and even all medicine), and not just neonatology. This provision should be corrected and the link corrected  also(1).

Author Response

Thank you for your comments and insight. We agree that given the high turnover of clinicians in the NICU setting which is unlikely to change given our current medical system setup, the communication around family context is crucial. While not the focus of our research, education is critical to improving our system.

In regards to your comments around the choice of our research methods, we determined that a descriptive qualitative approach would enable us to explore an in depth understanding of the problem and its nuances from the perspectives of parents and health care providers. A quantitative survey approach would imply that we already had this knowledge and felt that we knew exactly what to measure. This qualitative approach is strongly advocated for by quality improvement methodology (1) and well described in qualitative research methods (2-4). We have added a clarifying line highlighted below in paragraph two of methods to further expand on the purposive sampling that we used, as delineated by Patton (2).

“Using purposive sampling, we sought maximum variation by selecting families of varying cultural backgrounds and educational levels and clinicians of varying professions, cultural backgrounds and experience, were recruited (2). Recruitment was stopped for each group when thematic saturation was reached (5).”

As for the suggestion for stratification, we chose to use qualitative descriptive methods and move away from quantitative research methods such as stratification, to better understand each participant’s unique experience in depth as opposed to exploring the breadth of experiences based on a shared characteristic (2).  

Finally, we have broadened the statement around decision making and added the appropriate reference.

“In the late 20th century, decision-making in neonatology and in the wider medical field, shifted from a paternalistic model, in which physicians make decisions regarding care for patients and families, towards a model guided by recognition of patient autonomy (6, 7).”

Thank you again for your comments and we hope this explanation satisfies the reviewer and the editor. Please let us know should there be other issues to clarify.

Sincerely,

Maya Dahan on behalf of the authors

References:

  1. Robert G, Cornwell J, Locock L, Purushotham A, Sturmey G, Gager M. Patients and staff as codesigners of healthcare services. Bmj. 2015;350:g7714.
  2. Patton MQ. Qualitative Research and Evaluation Methods. Laughton CD, editor. United States of America: Sage Publications, Inc.; 2002.
  3. Braun V, Clarke V. Successful Qualitative Research: a practical guide for beginners. London, United Kingdom: SAGE Publications Inc. ; 2013.
  4. Hunter D, McCullum J, Howes D. Defining Exploratory-Descriptive Qualitative (EDQ) research and considering its application to healthcare. Journal of Nursing and Health Care. 2019;4.
  5. Hennink MM, Kaiser BN, Marconi VC. Code Saturation Versus Meaning Saturation: How Many Interviews Are Enough? Qual Health Res. 2017;27(4):591-608.
  6. Sullivan A, Cummings C. Historical Perspectives: Shared Decision Making in the NICU. Neoreviews. 2020;21(4):e217-e25.
  7. Emanuel EJ, Emanuel LL. Four models of the physician-patient relationship. Jama. 1992;267(16):2221-6.

Reviewer 2 Report

This is a well-written study concerning the family context and the participants-patient relationship in the pediatric context.

In my opinion, the manuscript have many issues, especially regarding rigor and methodology:

1) Do not use the acronym "NICU" without a previous clarification

2) please clearly express the method of the text analysis and the continent of the interviews

Author Response

Thank you for your comments and insights. We have added the backronym ‘Neonatal Intensive Care Unit’ in the abstract and the introduction.

In regards to your comment around the method of text analysis, please see paragraph three and four of method, copied below. We have added several phrases highlighted below to help further clarify our methods, the content of our interviews and how these have followed the guidance of thematic analysis, in accordance to Clarke and Braun (1).

“This study was reviewed and approved by the Research Ethics Board of Sunnybrook Health Sciences Center. Each participant consented for participation. Interviews occurred between August – December 2021. Interviews with the families were conducted by a physician (MD) and interviews with the clinicians were conducted by a research assistant with no prior relationship to the NICU or its’ staff. Interview guides were used to lead the interviews and were modified in response to iterative analysis. The interviews broadly focused on what participants shared about the FC, their experience of sharing the FC and the decisions around how and when to share information about the FC. Interviews were transcribed using artificial intelligence and corrected and anonymized by the lead researcher (MD).

A thematic analysis was performed on the interview transcripts (2). Two authors (MD+LR) read the first four transcripts for both groups (families and clinicians) and performed a preliminary analysis to generate a coding structure. The coding structure was then used by MD to code the remainder of the transcripts. The two lead researchers held regular analytic meetings following the steps for thematic analysis outlined by Braun and Clarke (1). Through coding and analysis, we identified, defined and named two important themes related to the experience of sharing the FC: the process of sharing the FC, and the impact of sharing the FC. The two themes were examined for how they were similar and different among and between the parents and clinician groups.”

If the reviewer and the editors have other specific information that they feel can be added to this description, please let us know what these are, and we would be happy to expand.

Thank you again for your comments and we hope this explanation satisfies the reviewer and the editor. Please let us know should there be other issues to clarify.

Sincerely,

Maya Dahan on behalf of all the authors

References:

  1. Braun V, Clarke V. Successful Qualitative Research: a practical guide for beginners. London, United Kingdom: SAGE Publications Inc. ; 2013.
  2. Terry GH, Nikki; Clarke, Victoria; Braun, Virginia. Thematic Analysis. In: Willig CR, Wendy Stainton, editor. The SAGE Handbook of Qualitative Research in Psychology. 55 City Road, London: SAGE Publications Ltd; 2017.

Reviewer 3 Report

The idea is awesome, the concept is new, at least for myself. It is well designed, well described, well discussed.Nice idea for improving daily practice. 

Author Response

Thank you for feedback and comments. We really appreciate you seeing how this study design really allows us to explore this novel concept and allows for future opportunity to really improve the care we provide.

Sincerely,

Maya Dahan on behalf of the authors

Round 2

Reviewer 1 Report

No new recommendations

Author Response

Dear Reviewer 1,

Thank you so much for your review. 

Sincerely,

Maya Dahan

Reviewer 2 Report

Dear authors,

As you know, reproducibility is one of the fundamental determinants of quality in research. In my opinion, the clear description of the interview process cannot be avoided.

For this reason, I will recommend rejection

Author Response

Dear Reviewer 2,

Thank you for your review. Our study methodology follows the COREQ Checklist which outlines the quality criteria for qualitative research, please see attached. We hope this helps address your concerns.

Sincerely,

Maya Dahan
